# The Performance, Physiology and Morphology of Female and Male Olympic-Distance Triathletes

**DOI:** 10.3390/healthcare10050797

**Published:** 2022-04-25

**Authors:** Paulo J. Puccinelli, Claudio A. B. de Lira, Rodrigo L. Vancini, Pantelis T. Nikolaidis, Beat Knechtle, Thomas Rosemann, Marilia S. Andrade

**Affiliations:** 1Programa de Pós-Graduação em Medicina Translacional, Department of Physiology, Federal University of São Paulo, São Paulo 04021-001, Brazil; paulopuccinelli@hotmail.com (P.J.P.); marilia1707@gmail.com (M.S.A.); 2Human and Exercise Physiology Division, Faculty of Physical Education and Dance, Federal University of Goiás, Goiânia 74690-900, Brazil; andre.claudio@gmail.com; 3Center of Physical Education and Sports, Federal University of Espírito Santo, Vitória 29075-910, Brazil; rodrigoluizvancini@gmail.com; 4School of Health and Caring Sciences, University of West Attica, 12243 Athens, Greece; pademil@hotmail.com; 5Medbase St. Gallen Am Vadianplatz, St. Gallen and Institute of Primary Care, 9100 St. Gallen, Switzerland; 6Institute of Primary Care, University of Zurich, 8091 Zurich, Switzerland; thomas.rosemann@usz.ch

**Keywords:** triathlon, sports medicine, sports physiology, female athlete, VO_2_ max

## Abstract

Sex differences in triathlon performance have been decreasing in recent decades and little information is available to explain it. Thirty-nine male and eighteen female amateur triathletes were evaluated for fat mass, lean mass, maximal oxygen uptake (VO_2_ max), ventilatory threshold (VT), respiratory compensation point (RCP), and performance in a national Olympic triathlon race. Female athletes presented higher fat mass (*p* = 0.02, *d* = 0.84, power = 0.78) and lower lean mass (*p* < 0.01, *d* = 3.11, power = 0.99). VO_2_ max (*p* < 0.01, *d* = 1.46, power = 0.99), maximal aerobic velocity (MAV) (*p* < 0.01, *d* = 2.05, power = 0.99), velocities in VT (*p* < 0.01, *d* = 1.26, power = 0.97), and RCP (*p* < 0.01, *d* = 1.53, power = 0.99) were significantly worse in the female group. VT (%VO_2_ max) (*p* = 0.012, *d* = 0.73, power = 0.58) and RCP (%VO_2_ max) (*p* = 0.005, *d* = 0.85, power = 0.89) were higher in the female group. Female athletes presented lower VO_2_ max value, lower lean mass, and higher fat mass. However, females presented higher values of aerobic endurance (%VO_2_ max), which can attenuate sex differences in triathlon performance. Coaches and athletes should consider that female athletes can maintain a higher percentage of MAV values than males during the running split to prescribe individual training.

## 1. Introduction

The participation of women in amateur and elite endurance sports events, including triathlon, has increased and their performance has improved during the last three decades [1,2,3,4,5,6]. Factors that are possibly associated with the increasing participation of women are the acceptance of female athletes in society, the importance of regular physical activity for the prevention and treatment of noncommunicable diseases, and the feeling of well-being that comes from a more active lifestyle [7].

Sex differences in triathlon performance seem to be decreasing, and currently vary between 12 and 18% [8,9], which seems to be influenced by distance, the level of competition, and the participation of the athletes [10,11].

Longer triathlon events, such as the Ironman (3.8 km of swimming, 180 km of cycling, and 42.2 km of running), or ultra-triathlons such as the Double Iron ultra-triathlon (7.6 km of swimming, 360 km of cycling, and 84.4 km of running) seem to be associated with decreased sex differences in performance, compared to shorter triathlon distances [12]. In addition, a lower tendency to sex difference was observed for elite athletes when compared to amateur athletes [12].

Some morphological sex differences related to body composition, such as lower fat mass percentage and higher muscle mass in the male sex [13,14,15], seem to be associated with better male performance [16].

Regarding the physiological factors that influence endurance performance, maximal oxygen uptake (VO_2_ max), ventilatory threshold (VT), and running economy (RE) are variables commonly investigated to predict aerobic performance [17].

In terms of VO_2_ max values, average values for females are approximately 75% of the values for males [18]. However, among athletes, these differences may be lower [19]. Lower blood hemoglobin concentrations typically found in women, as well as lower red cell mass and hematocrit level, which result in lower arterial oxygen content (CaO_2_) and lower O_2_ delivery to muscles during exercise [20,21] are the main factors responsible for these VO_2_ max gender differences.

Differently from VO_2_ max, data about sex differences according to the %VO_2_ max at VT seems to be contradictory. Female athletes have 7 to 23% more type I muscle fibers than men [22,23,24]. This difference in muscle fiber composition means that women have a greater oxidation of fat [25] and faster oxygen consumption kinetics [26], which should directly impact the VT, since this is dependent on the oxidative capacity of the muscles during exercise [27]. In addition, female athletes also have a higher rate of mitochondrial respiration [28]. These differences impact muscle metabolism, making women more apt to resynthesize ATP through the oxidative metabolism. Considering these sex differences, higher VT could be expected for female athletes; however, literature data show conflicting results. [17,18,19,20]. Moreover, as for VO_2_ max, there are very few data for VT in Olympic-distance triathletes [19].

Women’s participation in amateur triathlon events has increased in recent years. As a result, women’s performance has also improved, and sex differences have decreased [19]. So far, it is not possible to define whether the difference is associated with training volume or physiological limitations. Therefore, understanding physiological differences between the sexes can help clarify this issue.

Considering the importance of understanding the differences between sexes in endurance sports performance and the lack of data regarding both Olympic-distance triathletes and amateur athletes, we compared the physiological and morphological characteristics of male and female amateur triathletes of the same mean age who competed in an Olympic-distance event. Better knowledge about gender differences and female characteristics can explain the narrowing performance gap between the sexes of amateur triathlon athletes, and may help women reach their best performance.

We hypothesized that male triathletes would present higher VO_2_ max, higher lean mass, and lower fat mass than female triathletes, but that there would be no sex differences according to VT. Because of their higher VO_2_ max levels and better body composition, we hypothesized that men would present lower overall race time and split times than female triathletes in the Olympic triathlon race.

## 2. Materials and Methods

### 2.1. Ethical Approval

All experimental procedures were approved by the Human Research Ethics Committee of the Federal University of Sao Paulo (approval number 1659697) and conformed to the principles outlined in the Declaration of Helsinki. The study was conducted in accordance with recognized ethical standards and national/international laws. After receiving instructions regarding the experimental procedures, their possible risks and benefits, the objectives and justification of the research, and the principles of respect for persons involved, which encompassed a guarantee of privacy, confidentiality, and anonymity rights, the athletes signed the consent form.

### 2.2. Participants

Ninety-three athletes who had applied for Olympic-distance triathlon races accepted an invitation to participate in the study. However, thirty-six did not meet the inclusion criteria. Therefore, fifty-seven athletes participated in the study.

The inclusion criteria to participate in the study included having participated in at least one Olympic-distance triathlon race, with at least one year of triathlon practice. The exclusion criteria included having no medical approval for maximum effort, being pregnant, having acute pain in the lower limbs, edema, or not finishing the race.

The main reasons for the thirty-six exclusions were giving up on participating in the Olympic-distance triathlon race (*n* = 21), not finishing the race (*n* = 6), having injuries during the training period (*n* = 4), absence on the day scheduled for laboratory evaluations (*n* = 4), and one woman got pregnant.

Characterization of the sample according to the age and training habits are presented in Table 1.

As the number of female athletes who participated in the study was smaller than the number of male athletes, the power of the statistical analysis is shown with the *p*-value. This was employed to identify the possible lack of statistical difference between the groups due to the small sample size.

### 2.3. Procedures

Each participant reported to the laboratory for one day, in which they answered a questionnaire about training habits. Afterwards, anthropometric data measurement and a cardiorespiratory maximal test on a treadmill were performed. The organizing committee of the races provided the overall triathlon race time and split times. Thirty-nine male and six female amateur triathletes participated in the same race.

### 2.4. Assessments

#### 2.4.1. Questionnaires

The athletes answered a questionnaire about training habits with the four following questions: (1) How many years have you been practicing triathlon? (2) How many hours a week do you train swimming? (3) How many hours a week do you train cycling? (4) How many hours a week do you train running?

#### 2.4.2. Body Composition and Anthropometry

The height and body mass of the participants were assessed using a calibrated stadiometer and were measured to the nearest 0.1 kg and 0.1 cm, respectively. Dual energy X-ray absorptiometry (DXA, software version 12.3, Lunar DPX, GE Healthcare, Madison, WI, USA) was used to assess body composition (lean and fat mass). Athletes were instructed to follow their normal ad libitum hydration habits. They were evaluated after bladder voiding; no fasting or other limitations on their usual activities were implemented [29]. This method has been previously demonstrated as a reliable technique for body composition assessments [30,31].

#### 2.4.3. Cardiorespiratory Maximal Test on a Treadmill

Cardiopulmonary exercise testing (CPET) was conducted on a motorized treadmill (Inbrasport, ATL, Porto Alegre, Brazil) using a computer-based metabolic analyzer (Quark, Cosmed, Italy). The calibration procedure was performed prior to each test, according to the manufacturer’s guidelines. CPET was used to measure VO_2_ max, VT, respiratory compensation point (RCP), and maximal aerobic velocity (MAV). The VO_2_ max was determined as the stabilization of VO_2_ (increase lower than 2.1 mL·kg^−1^·min^−1^) even after increasing the treadmill velocity during the last stage of the CPET [32]. All the volunteers reached VO_2_ max. VT was determined based on the following criteria: an increase in the ventilatory equivalent for oxygen without an increase in the ventilatory equivalent for carbon dioxide, and an increase in the partial pressure of exhaled oxygen. RCP was determined based on the increase in the ventilatory equivalent for carbon dioxide and the decrease in the partial pressure of exhaled carbon dioxide [33]. VT and RCP were determined separately by two experienced investigators; a third investigator was asked in cases of discordance. MAV was determined as the minimal velocity eliciting the VO_2_ max [34]. The percentage of MAV that the athlete maintained during the running split was also calculated.

Athletes warmed up for 4 min at 10 km·h^−1^ (males) and 9 km·h^−1^ (females). After the warm-up period, the running velocity was increased by 1 km·h^−1^ every minute until voluntary exhaustion [35]. The entire CPET lasted between 8 and 12 min and treadmill grade was set at 1% to simulate the energetic cost of outdoor running [36]. The heart rate was measured by a monitor (Ambit 2S, Suunto, Finland) throughout the entire test, and perceived exertion was rated according to the Borg scale (a 10-point scale) [37].

### 2.5. Statistical Analysis

Data are presented as the mean and the standard deviations. All variables presented normal distribution and homogeneous variability according to the Shapiro–Wilk and Levene tests, respectively. In order to compare the triathlon race times and morphological and physiological characteristics of the sexes, Welch’s unequal variances t-test was used. This test was chosen because it is more reliable when the two samples have unequal sample sizes [38]. The measures of the effect size for differences between sexes were determined by calculating the mean difference between the two sexes, and then dividing the result by the pooled standard deviation. Calculating effect sizes, the magnitude of any change was judged according to the following criteria: *d* = 0.2 was considered a “small” effect size; 0.5 represented a “medium” effect size; and 0.8 a represented “large” effect size [39]. Considering that the study had a convenience sample, the power of all between-sex comparisons were calculated. Power analysis was performed using G*Power software [40]. The power of the tests varied from 0 to 1. Usually, researchers use 0.80 as the power level of the test [41]. Therefore, the same values were considered in this study to interpret the results. The level of significance was set at *p* < 0.05.

## 3. Results

Female athletes presented significantly lower body mass (*p* < 0.01, d = 2.00, power = 0.99) and height (*p* < 0.01, d = 1.80, power = 0.99) than male athletes. There was no difference in mean age between the groups (*p* = 0.21, d = 0.35, power = 0.65). Overall race time and split times were compared for sexes who participated in the same triathlon event. Regarding performance, female athletes presented higher race times for swimming (+11%), cycling (+7.5%), running (+7%), and overall race time (+8%). According to morphologic characteristics, male athletes presented higher lean body mass (kg) (*p* < 0.01, d = 3.11, power = 0.99). According to fat mass distribution, the percentage of trunk fat mass was not different between sexes (*p* = 0.522, *d* = 0.17, power = 0.73), nor was the percentage of android fat mass (*p* = 0.921, *d* = 0.02, power = 0.74), but the percentage of gynoid fat mass was higher in female athletes (*p* < 0.01, *d* = 1.37, power = 0.98). VO_2_ max (*p* < 0.01, *d* = 1.46, power = 0.99), MAV (*p* < 0.01, *d* = 2.05, power = 0.99), and velocities associated with VT (*p* = 0.02, *d* = 1.26, power = 0.97) and RCP (*p* < 0.01, *d* = 1.53, power = 0.99) were significantly higher in the male group. %VO_2_ max at VT (*p* = 0.012, *d* = 0.73, power = 0.58) and %VO_2_ max at RCP (*p* = 0.005, *d* = 0.85, power = 0.89) were higher in the female group. During the running split, female athletes were running at a higher percentage of MAV (75 ± 8%) than males (62 ± 6%) (*p* < 0.01, *d* = 1.83, power = 0.99) (Table 2).

## 4. Discussion

The primary aim of this study was to compare the sex differences of amateur Olympic-distance triathletes in relation to performance and physiological and morphological characteristics. The main findings were that: (i) the sex differences in performance were 8.0% for overall race time, 11% for swimming, 7.5% for cycling, and 7% for running; (ii) female athletes presented a lower VO_2_ max and a higher %VO_2_ max at VT and RCP than male athletes; (iii) female athletes presented lower lean mass than males; and (iv) female athletes presented higher total fat mass and gynoid fat mass than males, but the same android and trunk fat masses.

The sex differences in 1.5 km of swimming, 40 km of cycling, 10 km of running, and overall race time were 11.0, 7.5, 7.0, and 8.0%, respectively. Higher sex differences were previously shown for the top 10 athletes of each age group of the World Championship from 2009 to 2011, with a 13.3% performance difference in swimming, 10.7% difference in cycling, 7.5% difference in running, and 12% difference in overall race time [42]. Higher sex difference between the top five athletes from the “Zurich triathlon”, which occurs in Zurich, Switzerland, in each category have also been shown (18.5% in swimming, 15.5% in cycling, 18.5% in running, and 17.1% in overall race time) [6]. Therefore, it is evident that the sex differences in a given performance depend on the race level (world, national or regional championship). In the present study, minor differences were found between the sexes; however, only amateur athletes were studied, which differs from the studies mentioned above that evaluated elite athletes.

As expected, female athletes presented lower VO_2_ max and MAV values (49.5 ± 7.8 mL·kg^−1^·min^−1^ and 14.6 ± 1.7 km·h^−1^, respectively) than male athletes (59.9 ± 6.3 mL·kg^−1^·min^−1^ and 17.8 ± 1.0 km·h^−1^, respectively), showing a sex difference of approximately 19%. Similar sex difference have previously been shown for elite younger triathletes, reporting 20% lower values for females than for males (56.1 and 67.9 mL·kg^−1^·min^−1^) [43]. However, VO_2_ max values for ultra-endurance triathletes seem to be more similar between the sexes (68.8 and 65.9 mL·kg^−1^·min^−1^ for males and females, respectively), evidencing a sex difference of only 4.4% [44].

Besides maximal capacity for oxygen uptake, endurance performance also depends on VT. It has been suggested that 70% of success in endurance running depends on these physiological parameters [17]. An important new finding from the present study is that the female athletes presented higher values for VT (78.7 ± 6.1% for females and 74.4 ± 5.6% of VO_2_ max for males) and RCP (91.2 ± 4.1% for females and 87.5 ± 4.6% VO_2_ max for males) than male athletes. In addition, female athletes maintained a velocity corresponding to 75% of the MAV during the running split, which is higher than the value for males, who maintained 62% of their MAV [34]. The VT is limited by the peripheral conditions (i.e., mitochondrial volume, capillary density, oxidative enzyme capacity) [45,46]. Considering this context, females present different metabolic (greater proportional area of type I fibers [22,23,24], greater whole-muscle oxidative capacity [26], and greater mitochondrial oxidative function [28]), contractile (Ca^2+^ transients were smaller in magnitude and longer in duration in females [47]), and hemodynamic (greater vasodilatory responses of the arteries to muscles and higher density of capillaries per unit of skeletal muscle [22]) properties of skeletal muscles than males, favoring ATP resynthesis from oxidative phosphorylation during exercise [48,49], which could contribute to a higher VT.

Triathlon performance is also associated with body composition [16,50]. In this study, female athletes presented lower lean mass than males and higher total fat mass and gynoid fat mass percentage. The android and trunk body mass did not differ between the sexes. Moreover, fat mass values for both sexes were higher than those reported for elite athletes (<13% for female and <5% for males) [43]. Therefore, female body composition seems to be disadvantageous for athletic performance [13,14,51].

Regarding the limitations of this study, the test measurements cited were performed only on a treadmill. Thus, as the physiologic characteristics were only measured during a running activity using a treadmill, it would be very interesting to identify sex differences with the same measurements in tests performed during cycling or swimming activities. The inclusion of amateur athletes rather than elite athletes was another study limitation. Furthermore, this was a cross-sectional study, which prevented us from the studying the performance difference between sexes over time. Considering the increased popularity of Olympic-distance triathlon, especially among women, who were underrepresented in this sport until recently [52], the findings of the present study have practical applications for training monitoring. Strength and conditioning coaches working with triathletes might develop separate exercise programs for each sex.

Thus, an awareness of physiological sex differences related to performance would help coaches to prescribe sex-tailored training. In this context, the main finding from the present study was that the female athletes presented higher values of aerobic endurance (%VO_2_ max) than male athletes. These findings suggest that female athletes can maintain a higher percentage of MAV values than males during the running split; therefore, coaches could consider these findings to prescribe individual training.

## 5. Conclusions

In summary, female athletes present lower VO_2_ max and lean mass, and higher fat mass. However, they present higher values of aerobic endurance (%VO_2_ max), which can attenuate sex differences in triathlon performance. However, the sex differences in VT require further investigation, as there are few data about this variable in the literature.

## Figures and Tables

**Table 1 healthcare-10-00797-t001:** Characteristics of participants.

	Male Triathletes (*n* = 39)	Female Triathletes (*n* = 18)	*p* Value
Age (years)	38.8 ± 6.9	41.3 ± 6.68.4	0.210
Triathlon experience (years)	2.7 ± 1.7	3.3 ± 1.6	0.232
Training per week (hours)	13.2 ± 4.1	14.4 ± 3.5	0.287

Data are presented as mean ± standard deviations.

**Table 2 healthcare-10-00797-t002:** Descriptive characteristics of the triathletes and comparison between the sexes.

	Male (*n* = 39)	Female (*n* = 18)	*p* Value	*d* Value	Power (1-Beta)
**Anthropometric profile**					
Age (years)	38.9 ± 6.9	41.3 ± 6.6	0.21	0.35	0.65
Body mass (kg)	74.3 ± 8.8 *	59.5 ± 5.6	<0.01	2.00	0.99
Height (cm)	174.8 ± 6.5 *	164.5 ± 4.8	<0.01	1.80	0.99
Fat mass (%)	16.8 ± 5.6 *	23.2 ± 9.2	0.02	0.84	0.78
Lean Mass (kg)	59.0 ± 5.7 *	43.0 ± 4.5	<0.01	3.11	0.99
Trunk fat mass (%)	19.8 ± 6.8	21.3 ± 10.2	0.52	0.17	0.73
Android fat mass (%)	22.7 ± 8.6	22.4 ± 12.0	0.92	0.02	0.74
Gynoid fat mass (%)	21.9 ± 6.2 *	33.2 ± 9.8	<0.01	1.37	0.98
**Maximal graded exercise test**					
VO_2_ max (ml·kg^−1^·min^−1^)	59.9 ± 6.3 *	49.5 ± 7.8	<0.01	1.46	0.99
VT (%VO_2_ max)	74.4 ± 5.6 *	78.7 ± 6.1	0.01	0.73	0.58
Velocity at VT (km·h^−1^)	12.4 ± 1.4 *	10.5 ± 1.6	<0.01	1.26	0.97
RCP (%VO_2_ max)	87.5 ± 4.6 *	91.2 ± 4.1	0.01	0.85	0.89
Velocity at RCP (km·h^−1^)	14.8 ± 1.5 *	12.5 ± 1.5	<0.01	1.53	0.99
MAV (km·h^−1^)	17.8 ± 1.4 *	14.6 ± 1.7	<0.01	2.05	0.99
**Running split**					
%MAV	62 ± 6 *	75 ± 8	<0.01	1.83	0.99
Velocity (km·h^−1^)	11.0 ± 1.0 *	11.0 ± 1.8	0.99	0.00	0.99

Data are presented as mean ± standard deviations. *d* value: Effect size (Cohen’s D). VO_2_ max: maximal oxygen uptake. VT: ventilatory threshold. RCP: respiratory compensation point (RCP). MAV: maximal aerobic velocity. * significant difference between sexes (*p* < 0.05).

## Data Availability

Data supporting reported results can be asked to corresponding author.

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
