# Peer review of "The Performance, Physiology and Morphology of Female and Male Olympic-Distance Triathletes"

_healthcare, 2022, doi:10.3390/healthcare10050797_

Round 1

Reviewer 1 Report

After second revison I have no further suggestions how to improve the paper. From my point of view it could be considered for publication.

Author Response

Reviewer #1

After second revison I have no further suggestions how to improve the paper. From my point of view it could be considered for publication

Answer: Thank you about your positive feedback.

Reviewer 2 Report

The main aim of this study was to compare sex differences of amateur Olympic distance triathletes in relation to performance and physiological and morphological characteristics. Regarding the authors, I would like to congratulate and thank them for their effort and motivation involved in this research study. The presentation of the research is well documented, with a scientific basis and respects the latest standards regarding the highest level scientific publications. The methodology was chosen correctly. The conclusions support and result from the research and open new directions for future research. The submitted work is interesting and essentially exhausts the subject under discussion.

Apart from the fact that informed consent was obtained from all participants in the study (which is an obligation and for which I thank you), what I miss is whether the athletes, when answering the questionnaire, were informed of any other guidelines, in line with the recommendations of the Publication Manual of the American Psychological Association, such as, for example, the principles of privacy, confidentiality, anonymity. I would kindly ask you to expand the article with such information.

Supplementing the article with the above-mentioned issue is the only point that I think needs to be added. I keep my fingers crossed for the final success of the publication, which in my opinion is really good.

Author Response

Reviewer #2

The main aim of this study was to compare sex differences of amateur Olympic distance triathletes in relation to performance and physiological and morphological characteristics. Regarding the authors, I would like to congratulate and thank them for their effort and motivation involved in this research study. The presentation of the research is well documented, with a scientific basis and respects the latest standards regarding the highest level scientific publications. The methodology was chosen correctly. The conclusions support and result from the research and open new directions for future research. The submitted work is interesting and essentially exhausts the subject under discussion.

Answer: Thank you about your positive feedback.

Apart from the fact that informed consent was obtained from all participants in the study (which is an obligation and for which I thank you), what I miss is whether the athletes, when answering the questionnaire, were informed of any other guidelines, in line with the recommendations of the Publication Manual of the American Psychological Association, such as, for example, the principles of privacy, confidentiality, anonymity. I would kindly ask you to expand the article with such information.

Answer: Thank you about your constructive commentary. The following sentence has been added in the manuscript to clarify this point: “After receiving instructions regarding the experimental procedures, their possible risks and benefits, the objectives and justification of the research, as well as the principle of respect for persons, which encompasses the guarantee of privacy, confidentiality, anonymity rights, athletes signed the consent form.” Please let us know if the inclusion of this sentence do not resolve your doubts.

Supplementing the article with the above-mentioned issue is the only point that I think needs to be added. I keep my fingers crossed for the final success of the publication, which in my opinion is really good.

Answer: Thank you about your positive feedback.

This manuscript is a resubmission of an earlier submission. The following is a list of the peer review reports and author responses from that submission.

Round 1

Reviewer 1 Report

Congratulations on the study, thinking of individualizing the protocols and training, it is very interesting to already separate by gender characteristics, however we have the need for sampling and methodological adjustments, namely:
a) lack of data referring to the strength levels of the athletes would be interesting to highlight some points;
b) it is important to have adequate numbers of participants for the study.

Obs.: We have a change to make on line 216 - error in subtitle vt at

Author Response

Reviewer #1

Congratulations on the study, thinking of individualizing the protocols and training, it is very interesting to already separate by gender characteristics, however we have the need for sampling and methodological adjustments, namely:
a) lack of data referring to the strength levels of the athletes would be interesting to highlight some points;

Answer: Thank you about your positive feedback. The aim of the present study was to compare the anthropometric profile (weight, height and body composition) and the physiological determinants of aerobic performance (VO2max, % VO2max that could be sustained and running economy). We didn't aim to compare strength. Therefore, unfortunately, we don't have this data.

  1. b) it is important to have adequate numbers of participants for the study.

Answer: Despite the number of male and female athletes was different, when analyzing the differences in morphological and physiological characteristics, the power of analyzes was higher than 0.80, which ensures sufficient security to draw conclusions resulting from the results of statistical analyzes. Therefore, we conclude that the sample size was enough to support the study conclusions.

Obs.: We have a change to make on line 216 - error in subtitle vt at

Answer: Thank you for calling your attention to this point. The mistake has been corrected.

Reviewer 2 Report

The present study aimed to investigate the sex differences in triathlon performance and the mechanism behind the difference. Thirty-nine male and six female amateur triathletes were recruited and certain indicators were assessed, including fat mass, lean mass, maximal oxygen uptake (VO2max), ventilatory threshold (AT), respiratory compensation point (RCP), running economy (RE), and performance in a national Olympic triathlon race. Significant differences were found in these indicators. The authors concluded that higher values of aerobic endurance in females may attenuate sex differences in triathlon performance. The topic is interesting. However, there are several concerns for the authors' further considerations.

  1. The rationale of the present study is not very clear. Actually, sex differences in the morphology data and physiological data have been well documented in previous studies. It has been suggested that these differences may be, at least partly, responsible for the differences in exercise performance. Why do the authors believe that these sex differences may explain the changes in endurance performance? Are there any studies that supported the decreased sex differences in these data?
  2. Another concern is the sample size. Only 6 female participants were included. Will this affect the results significantly?
  3. The results are expected. What are the new findings of the present study? These should be highlighted and further discussed.

Author Response

Reviewer # 2

The present study aimed to investigate the sex differences in triathlon performance and the mechanism behind the difference. Thirty-nine male and six female amateur triathletes were recruited and certain indicators were assessed, including fat mass, lean mass, maximal oxygen uptake (VO2max), ventilatory threshold (AT), respiratory compensation point (RCP), running economy (RE), and performance in a national Olympic triathlon race. Significant differences were found in these indicators. The authors concluded that higher values of aerobic endurance in females may attenuate sex differences in triathlon performance. The topic is interesting. However, there are several concerns for the authors' further considerations.

  1. The rationale of the present study is not very clear. Actually, sex differences in the morphology data and physiological data have been well documented in previous studies. It has been suggested that these differences may be, at least partly, responsible for the differences in exercise performance. Why do the authors believe that these sex differences may explain the changes in endurance performance? Are there any studies that supported the decreased sex differences in these data?

Answer: Thank you about your constructive comment. Female athletes have 7 to 23% more type-I muscle fibers than men and this difference in the composition of muscle fibers makes women present greater oxidation of fat (Tarnopolsky, 2008) and faster oxygen consumption kinetics. In addition, female athletes also have a higher rate of mitochondrial respiration. These differences impact muscle metabolism, making women more apt to resynthesize ATP through oxidative metabolism, which should directly impact the ventilatory threshold and, therefore, the endurance performance. This explanation was included in the introduction section. There are several studies that show differences between the sexes of these physiological factors that should produce differences in ventilatory thresholds (Simoneau and Bouchard, 1989; Miller et al., 1993; Staron et al., 2000; Roepstorff et al., 2006; Tarnopolsky, 2008; Cardinale et al., 2018).

  1. Another concern is the sample size. Only 6 female participants were included. Will this affect the results significantly?

Answer: Despite the number of male and female athletes was different, when analyzing the differences in morphological and physiological characteristics, the power of analyzes was higher than 0.80, which ensures sufficient security to draw conclusions resulting from the results of statistical analyzes. Therefore, we conclude that the sample size was enough to support the study conclusions.

  1. The results are expected. What are the new findings of the present study? These should be highlighted and further discussed.

Answer: Thank you about your constructive comments. The discussion section has been rewritten in order to clarify the main and new findings. The higher ventilatory threshold presented by the female athletes was and an important new finding from the present study, which also can be used by coaches to prescribe individualized training sections.   

Reviewer 3 Report

"Although some authors af-255 firm that there are no reason to expect higher adaptation to peripheral training in female 256 than in male athletes [17,39], females present different metabolic (greater proportional 257 area of type I fibers, greater whole-muscle oxidative capacity and greater mitochondrial 258 oxidative function), contractile (slower Ca2+ kinetics and slower shortening and relaxation 259 velocities) and hemodynamic (greater vasodilatory responses of the arteries to muscles 260 and higher density of capillaries per unit of skeletal muscle) properties of skeletal muscles 261 from the males, favoring the ATP resynthesis from oxidative phosphorylation during ex-262 ercise, which may possibly be contributing to a higher ventilatory threshold."

Paper is clearly written, with adequate methods, description of the results and disscussion.  One of the findings stated in the paper is the higher adaptation to peripheral training in women. Since this is one of the new findings of the paper, and authors cite literature where there are no reasons to expect higher adaptation to peripheral training in female than in male athletes the problem is that afterwards they stated that there could be are several adaptations at the pheriphery. However that statment is not supported by other literature (it should be) or by the results obtained in the study. uthors should find available literature for this. If not that explanation could be misleading.

Author Response

Reviewer #3

"Although some authors affirm that there are no reason to expect higher adaptation to peripheral training in female  than in male athletes [17,39], females present different metabolic (greater proportional area of type I fibers, greater whole-muscle oxidative capacity and greater mitochondrial oxidative function), contractile (slower Ca2+ kinetics and slower shortening and relaxation velocities) and hemodynamic (greater vasodilatory responses of the arteries to muscles and higher density of capillaries per unit of skeletal muscle) properties of skeletal muscles from the males, favoring the ATP resynthesis from oxidative phosphorylation during exercise, which may possibly be contributing to a higher ventilatory threshold."

Paper is clearly written, with adequate methods, description of the results and disscussion.  One of the findings stated in the paper is the higher adaptation to peripheral training in women. Since this is one of the new findings of the paper, and authors cite literature where there are no reasons to expect higher adaptation to peripheral training in female than in male athletes the problem is that afterwards they stated that there could be are several adaptations at the pheriphery. However that statment is not supported by other literature (it should be) or by the results obtained in the study. Authors should find available literature for this. If not that explanation could be misleading.

Answer: Thank you about your constructive comments. The sentence has been rewritten and new references had been included.

Round 2

Reviewer 2 Report

Thanks for the authors' effort in revising their paper. The quality of the paper has improved. I have one more minor concern:

One rationale of the present study was that the authors believed that sex differences in triathlon performance were decreasing, and it was hypothesized that sex differences in physiological factors might be one possible reason. However, according to the results of the present study, we could not observe the decreased sex differences in performance. This may be one limitation and should be further discussed. Additionally, the inclusion of amateur athletes but not elite athletes was definitely one limitation, and should be mentioned in the discussion/limitation section.

Author Response

Thanks for the authors' effort in revising their paper. The quality of the paper has improved. I have one more minor concern:

One rationale of the present study was that the authors believed that sex differences in triathlon performance were decreasing, and it was hypothesized that sex differences in physiological factors might be one possible reason. However, according to the results of the present study, we could not observe the decreased sex differences in performance. This may be one limitation and should be further discussed. Additionally, the inclusion of amateur athletes but not elite athletes was definitely one limitation, and should be mentioned in the discussion/limitation section.

Answer: Thank you about your positive and constructive comment. We agree with the reviewer. The fact that only amateur athletes were studied was included in the discussion section. In addition, despite the evident physiological and morphological differences between the sexes that were evidenced, it is not possible to attribute the decrease in the performance difference between the sexes to the decrease in the difference of the physiological variables, since it is a cross-sectional study. This consideration has also been included in the discussion/limitation section.